# Multiple Levels of Triggered Factors and the Obligated Requirement of Cell-to-Cell Movement in the Mutation Repair of Cucumber Mosaic Virus with Defects in the tRNA-like Structure

**DOI:** 10.3390/biology11071051

**Published:** 2022-07-13

**Authors:** Shanshan Liu, Jinze Mu, Chengming Yu, Guowei Geng, Chenyu Su, Xuefeng Yuan

**Affiliations:** Shandong Province Key Laboratory of Agricultural Microbiology, Department of Plant Pathology, College of Plant Protection, Shandong Agricultural University, Tai’an 271018, China; shansd1218@126.com (S.L.); mjz980119@163.com (J.M.); ycm2006.apple@163.com (C.Y.); guowgeng@163.com (G.G.); dinoscy@163.com (C.S.)

**Keywords:** mutation repair, tRNA-like structure, cucumber mosaic virus

## Abstract

**Simple Summary:**

Based on analysis of the tRNA-like structure (TLS) mutation in cucumber mosaic virus (CMV), mutation repair is correlated with several levels of triggered factors, including the dose of inoculation of virus mutants, the quantity effect on corresponding viral RNA, and the quality effect on corresponding viral RNA. All types of TLS mutation in different RNAs of CMV can be repaired at a low dose around the dilution end-point. At a high dose of inoculation, TLS mutations in RNA2 and RNA3, but not RNA1, can be repaired, which correlates with the relative quantity defect of RNA2 or the genome size defect of RNA3. In addition, all the above types of mutation repair necessarily require cell-to-cell movement, which presents the obligated effect of cell-to-cell movement on mutation repair.

**Abstract:**

Some debilitating mutations in RNA viruses are repairable; however, the triggering factors of mutation repair remain largely unknown. In this study, multiple triggering factors of mutation repair are identified based on genetic damage to the TLS in CMV. TLS mutations in different RNAs distinctively impact viral pathogenicity and present different types of mutation repair. RNA2 relative reduction level or RNA3 sequence change resulting from TLS mutation is correlated with a high rate of mutation repair, and the TLS mutation of RNA1 fails to be repaired at the high inoculum dose. However, the TLS mutation of RNA1 can be repaired at a low dose of inoculation, particularly around the dilution end-point or in the mixed inoculation with RNA2 having a pre-termination mutation of the 2b gene, an RNAi suppressor. Taken together, TLS mutations resulting in quality or quantity defects of the viral genome or TLS mutations at low doses around the dilution end-point are likely to be repaired. Different levels of TLS mutation repair necessarily require cell-to-cell movement, therefore implying its obligated effect on the evolution of low-fitness viruses and providing a new insight into Muller’s ratchet. This study provides important information on virus evolution and the application of mild viral vaccines.

## 1. Introduction

RNA viruses have evolved several strategies such as negative selection—which is also called purifying selection and is the most prevalent form of selection—and RNA recombination to maintain their genome integrity against mutations, owing to the infidelity of replicative machinery and adverse factors [1,2,3]. Negative selection can eliminate less-fit or lethal variants, while some debilitating mutations in RNA viruses can be repaired or remodeled by direct reversion or compensatory mutations, even in trans-complementation from other viruses [2,3]. RNA recombination plays an essential role in the genetic evolution and stability of all living organisms and is responsible for viral gene rearrangement, the repair of debilitating mutations, and the acquisition of non-self sequences [4]. Replicative systems of RNA viruses are typically error-prone, primarily due to the low fidelity of their RNA-dependent RNA polymerases (RdRp) and the lack (in the majority of cases) of proofreading mechanisms [3]. Genomes of well-adapted RNA viruses can maintain their identity based on the error-prone (infidelity) element of the replicative machinery [5,6]. However, the genomes of weak-adapted or debilitated RNA viruses could undergo changes during competition with their derivates due to the error-prone element of the replicative machinery, which may cause mutation repair or even lead to higher-fitness variants [3]. RNA recombination can shape the robustness and resilience of RNA viruses, which are two connected and conflicting aspects of viral evolution [3,4]. RNA recombination has accelerated the evolution and adaptation of positive-strand RNA viruses and the emergence of new viral variants [7].

RNA recombination can be divided into homologous, aberrant homologous, and non-homologous recombinations [8,9]. Two models have been proposed to interpret RNA recombination: the potential cleavage–religation of precursor RNAs for non-homologous recombinations, and the replicase-mediated copy-choice mechanism for both homologous and non-homologous recombinations [10,11,12,13], which is the most widely accepted recombination model. Replication-related RNA recombination is mediated by a copy-choice mechanism that occurs at specific crossover sites between different RNA templates from related viruses and host RNAs [4,14]. During the copy-choice mechanism, the specific crossover site between two RNA templates and an error-prone replicase leads to RNA recombination [4]. Template conversion is affected by a variety of factors such as host, environment, the degree of local sequence identity between RNA templates, transcription kinetics, and secondary structure in RNA [15,16,17,18]. The frequency of RNA recombination depends on mechanistic and/or ecological factors [4]. The local concentration of replication proteins in the viral replicase may act as a determinant in replication-related recombination processes [7]. A recent study has reported that the control elements of recombination frequency reside predominantly within the 2a gene of RNA2 in cucumber mosaic virus (CMV) [19]. Each RNA virus seems to be capable of recombining, as RNA recombination does not require complicated machinery [4]. RNA recombination promotes the evolution in RNA viruses and may cause mutation repair, debilitating viral genomes. However, the factors triggering or determining the frequency of mutation repair via RNA recombination have not been identified. Moreover, the identification requires information on deleterious mutations and specific crossover sites.

In this study, a multipartite CMV is used to study the role of RNA recombination on mutation repair. The CMV genome comprises three RNAs (RNA1, RNA2, and RNA3). RNA1 encodes 1a, a replication-associated protein; RNA2 encodes 2a (RNA-dependent RNA polymerase, RdRp) and 2b (a gene-silencing suppressor); and RNA3 encodes 3a (movement protein) and CP (coat protein). We express 2b using subgenomic RNA4A derived from RNA2 and CP using subgenomic RNA4 derived from RNA3 [20]. CMV RNAs have a similar 3′UTR containing a conserved tRNA-like structure (TLS), which is essential for replication and has some core cis-elements, such as the caacg loop [21]. We observe different frequencies of repair for identical mutations in TLSs of different CMV RNAs. The results suggest that, in addition to quality defects (sequence changes in RNA3) or quantity defects (the relative reduction level in RNA2) in the genome caused by TLS mutation, mutation repair could be triggered by low inoculum doses. In addition, TLS mutation repair requires cell-to-cell movement. These results provide new insights into viral evolution and mild vaccine application.

## 2. Materials and Methods

### 2.1. Plasmid Construction

All mutant plasmids were derived from pCB301-Fny1, pCB301-Fny2, and pCB301-Fny3, which contained the wild-type RNA1 (D00356), RNA2 (D00355), and RNA3 (D10538) sequences of the CMV_Fny_ isolate [22]. Two types of TLS mutants were constructed in RNA1, RNA2 or RNA3 of CMV_Fny_. Mutant T_m1_ of TLS was the replacement of the essential caacg loop in TLS with an 18 nt sequence ggatccactagtcccggg. Mutant T_m2_ of TLS was the insertion of a new 18 nt sequence ggatccactagtcccggg before the caacg loop in TLS. The mutant R2-2bPT was inserted into two stop codons (UAAUAG) and MCS (ggatccactagtcccggg) after position ^2661^G in RNA2 to cause the pre-termination of the 2b protein. The mutant R2-2bPT-1 and R2-2bPT-2 were inserted 100 bp (7600–7699) and 200 bp (7600–7799) fragments from TVBMV (EU734432) into the R2-2bPT through the MCS (ggatccactagtcccggg), respectively. Mutant R3-M5 was produced by the mutation of ^549^tatgattgt to ^549^gctgcatgc in the 3a protein, causing cell-to-cell movement defects [23].

All the above mutants were constructed with the plasmids pCB301-Fny1, pCB301-Fny2, or pCB301-Fny3 through oligo-mediated mutagenesis, which was performed by site mutagenesis mediated by primers [24]. Detailed information on the plasmids and primers is shown in Appendix A. All the mutants were confirmed by DNA sequencing.

### 2.2. Plasmid Transformation and Agroinfiltration

All plasmids were transformed into *Agrobacterium tumefaciens* GV3101 using the CaCl_2_-mediated freeze–thaw method, as described previously [25]. Single *A. tumefaciens* colonies were cultured in LB medium with 50 µg/mL kanamycin and 100 µg/mL rifampin at 28 °C until OD_600_ reached 1.0–2.0. After centrifugation, the pellets were resolved to an OD_600_ of 1.2 in an infiltration solution (10 mM MgCl_2_, 10 mM 2-(N-morpholino), and 0.15 mM acetosyringone). GV3101 containing plasmids of RNA1, RNA2, and RNA3, as well as the mutants, were equally mixed. The mixture was further diluted in a dilution assay based on the requirements with the infiltration solution. The mixture was incubated at 28 °C for 3 h and infiltrated into the 4th and 5th leaves of *N. benthamiana* at the 6–7 leaf stage. Plants were grown under conditions of a 16 h photoperiod and 8 h of darkness at 25 °C, as described previously [26]. Each treatment was repeated at least three times, and at least three plants were used in each treatment set.

### 2.3. Mutation Repair Assay

To evaluate the stability and repair of mutation sites in progeny RNAs, total RNA was extracted from inoculated and/or systematic leaves at 14 dpi after agroinfiltration. Subsequently, RT-PCR was performed using specific primers located upstream and downstream of the corresponding mutation sites. The RT-PCR products were purified and sanger sequencing was performed using specific primers. Sequencing results were analyzed to determine whether the mutation sites underwent mutation repair and maintained the mutation or if there was the co-existence of mutation repair and original mutation in the progeny RNAs, based on the sequencing figure. If there were overlapped peaks of original mutation sites in the sequencing figure, this suggested the co-existence of mutation repair and original mutation. Detailed information on the primers used to amplify the mutated regions is provided in Appendix A. For each treatment, viral progeny RNAs of 100 plants from five batches in the inoculation assay were analyzed by RT-PCR and DNA sequencing.

Based on the sequencing data of the RT-PCR products, the mutation repair frequency was calculated as follows:$$\mathrm{Frequency}\;\mathrm{of}\;\mathrm{mutation}\;\mathrm{repair}=\frac{\mathrm{number}\;\mathrm{of}\;\mathrm{plants}\;\mathrm{containing}\;\mathrm{mutation}\;\mathrm{repair}\;}{\mathrm{total}\;\mathrm{number}\;\mathrm{of}\;\mathrm{plants}\;\mathrm{infected}\;\mathrm{by}\;\mathrm{mutants}}\times 100\%$$

### 2.4. Northern Blot

Total RNA was extracted from systemic non-inoculated leaves 14 dpi after agroinfiltration, and Northern blotting was performed as previously described [27]. cDNA probes were labelled using random primers. Four types of cDNA probes were used to detect different genomic and corresponding subgenomic RNAs. cDNA probe 1 (cP1) corresponding to 2951–3150 of RNA1 was designed to exclusively detect RNA1. cDNA probe 2 (cP2), corresponding to 2548–2730 of RNA2, was designed to exclusively detect RNA2. cDNA probe 3 (cP3), corresponding to 1704–1910 of RNA3, was designed to exclusively detect RNA3. cDNA probe 4 (cP4), corresponding to the 3′ end of RNAs in CMV, was designed to simultaneously detect RNA1, RNA2, and RNA3, as well as corresponding subgenomic RNAs. Detailed information on the primers used to amplify the PCR products for preparing the probes is shown in Appendix A.

### 2.5. Statistical Analysis

The blot signals were analyzed with Quantity One software. Mean values of at least three independent experiments are shown and standard deviations (S.D.) are given. Duncan’s multiple range test, analysis of variance (ANOVA), and the independent sample LSD test were performed in DPS software. *p* < 0.05 denotes significant differences between comparisons.

## 3. Results

### 3.1. TLS Mutation in Different RNAs of CMV Showed Remarkable Differences in Repair Frequency, Mediated by Copy-Choice-Type RNA Recombination

In this study, the tRNA-like structure (TLS) was mutated to cause a potentially debilitating genome because TLS has been reported to be essential for the in vitro replication of CMV [21]. TLSs of RNA1, RNA2, and RNA3 were mutated by replacing the caacg loop with 18 nt sequences and inserting 18 nt sequences in the caacg loop (Figure 1A, Appendix A). Based on the structural alignment of TLS regions among wt RNAs, T_m1_, and T_m2_ mutants (Appendix A), the secondary structures of the TLS regions in wt RNA1, RNA2, and RNA3 presented conserved characteristics. In T_m1_ mutants, the replacement of the caacg loop with 18 nt changed the upper part of the “C” part in the TLS in a similar pattern as in RNA1, RNA2 and RNA3 (Appendix A). In T_m2_ mutants, the insertion of 18 nt before the caacg loop dramatically changed the secondary structure of the TLS regions in a similar pattern in the three RNAs (Appendix A). T_m1_- or T_m2_-type mutations in the TLS might cause a debilitating genome due to the change in caacg loop or the whole TLS. At 14 dpi of agroinfiltration at OD_600_ = 1.2, viral progeny RNAs from systemic leaves in different plants were performed by reverse-transcriptase PCR, and purified PCR products were directly sequenced to identify the characteristics of the mutation sites. For each treatment, RNA samples from 100 plants in 5 batches of the inoculation assay were tested. RNA from one plant was termed as a statistic unit. At 14 dpi of agroinfiltration at OD_600_ = 1.2, the two types of TLS mutations in RNA1 (R1-T_m1_ and R1-T_m2_) still existed in progeny RNAs and were not repaired. However, TLS mutations in RNA2 or RNA3 were found to have different repair frequencies in progeny RNAs, which repaired the original mutations to wild-type sequences (Figure 1B). The repair frequencies of this type of original mutation to wild type in R2-T_m1_, R2-T_m2_, R3-T_m1_, and R3-T_m2_ were 46%, 35%, 80%, and 67%, respectively (Figure 1B). Therefore, identical TLS mutations in RNA1, RNA2, and RNA3 of CMV had different repair characteristics, including the certainty and frequency of repair at high inoculum doses.

### 3.2. Quality or Quantity Defects of Genomic RNAs via TLS Mutation in RNA2 or RNA3 of CMV, Triggering the Repair Mechanism

To further identify the factors of mutation repair, the effect of different TLS mutations on CMV pathogenicity and RNA accumulation was analyzed. Mutants R1-T_m1_ and R1-T_m2_ had similar pathogenicity to wt and caused severe stunting (Figure 1C). Mutants R2-T_m1_ and R2-T_m2_ had weak pathogenicity, causing a little stunting (Figure 1C). Mutants R3-T_m1_ and R3-T_m2_ had a weaker pathogenicity and stunting effect than that of wt (Figure 1C). Therefore, identical TLS mutations in RNA1, RNA2, and RNA3 had different effects on CMV pathogenicity. When the TLS mutation in RNA2 or RNA3 was repaired, symptoms of the mutant precursor were still noticed (Figure 1D), which implies that mutation repair requires a process, not a sudden action. To identify why TLS mutations in RNA1, RNA2, and RNA3 had different effects on CMV pathogenicity, the characteristics of CMV RNAs in samples without direct TLS mutation repair were analyzed by Northern blot (Figure 1E,F). The results indicated that single TLS mutation in CMV RNAs did not affect the RNA accumulation of the CMV genome (Figure 1E,F). However, single TLS mutations in RNA1, RNA2, and RNA3 had different effects on specific RNA segments when the relative ratio between the RNAs was analyzed. The TLS mutation of R1-T_m1_ or R1-T_m2_ did not affect the synthesis of RNA1, RNA2, or RNA3 (Figure 1E,F). The R2-T_m1_ and R2-T_m2_ mutants decreased the relative accumulation of RNA2/RNA3 by 60% and 20%, respectively (Figure 1F). The TLS mutants of RNA1 and RNA3 slightly increased the relative synthesis of RNA2/RNA3 (Figure 1F). Mutants of R3-T_m1_ or R3-T_m2_ increased the size of RNA3 and its subgenomic RNA (RNA4) (Figure 1E,F). The size of RNA3 and RNA4 were normal in samples with the direct repair of TLS mutation in RNA3 (Figure 1G). Northern blot analysis of R1/R2/R3-T_m1_ and R1/R2/R3-T_m2_ showed that RNA3 and RNA4 components were longer than those of wild-type, RNA1, and RNA2 mutants (Figure 1F). Therefore, an extended sequence may have been contained in the 3’ half of RNA3 and RNA4. To identify the extended sequences, the region 1183–2216 of RNA3 was reverse-transcribed (Figure 2A). Products of two different sizes were observed for R1/R2/R3-T_m1_ and R1/R2/R3-T_m2_, and the larger products were purified and cloned into the pEASY-T5 Zero vector (Figure 2A). Subsequent sequencing using the universal primer M13F/M13R revealed approximately 300 bp of the extended sequences located at the proximal 3′end, which was derived from the 3′ end of RNA2 (Figure 2B and Appendix A). In the larger RNA3 of R1/R2/R3-T_m1_ and R1/R2/R3-T_m2_, the extended sequences were from sequences of position 2756–3050 and 2713–3050 of RNA2, respectively (Figure 2B and Appendix A). Although TLS is conserved in different RNAs of CMV, TLS mutations in RNA1, RNA2, and RNA3 had entirely different effects on the synthesis of the corresponding RNAs (Figure 1E,F and Figure 2). Taken together, the repair frequency of TLS mutations in different CMV RNAs may be correlated with the negative effect on the synthesis of the corresponding RNAs (Figure 1). Quality defects of larger RNA3/RNA4 size caused by TLS mutations (T_m1_ or T_m2_) in RNA3 triggered a high frequency of mutation repair (80% or 67%). Quantity defects in RNA2 synthesis caused by TLS mutations (T_m1_ or T_m2_) in RNA2 also triggered a relatively low frequency of mutation repair (46% or 35%). The TLS mutation in RNA1 did not cause a quantity or quality defect, which is correlated with the non-occurrence of mutation repair. The quality or quantity defects in the genome segment of CMV caused by TLS mutation were important trigger factors for mutation repair at the high inoculum doses.

Because of the conserved TLS region in CMV RNAs, RNA recombination may be a candidate mechanism to mediate the repair of TLS mutations. To identify this fact, identical TLS mutants of RNA1, RNA2, and RNA3 were mixed to infect plants simultaneously and mutation repair was analyzed (Figure 3). Mixed infection with R1-T_m1_/R2-T_m1_/R3-T_m1_ or R1-T_m2_/R2-T_m2_/R3-T_m2_ resulted in low pathogenicity and no stunting (Figure 3A), which was consistent with low virus accumulation in the Northern blot (Figure 3B). These two types of mixed infections did not show TLS mutation repair (Figure 3C). No repair of the TLS mutation during infection with R1-T_m1_/R2-T_m1_/R3-T_m1_ was due to the absence of wild-type TLS sequences. When infected with R1-T_m2_/R2-T_m2_/R3-T_m2_, wild-type caacg and insertion sequences were mixed, which may not have met the requirement of the wild-type sequences during copy-choice. Therefore, TLS mutation repair might be mediated by copy-choice-type RNA recombination.

### 3.3. Mutation Repair of Debilitating Genomes Could Be Triggered by Critical-Threshold Low Dose, Especially around Dilution End-Point

Although the TLS mutation of RNA1 failed to be repaired (Figure 1), it caused a debilitating TLS in RNA1. Can the debilitating TLS in RNA1 be repaired under other special conditions, such as mixed infection with other types of mutants that cause genome defects, such as low RNA accumulation? The pre-termination mutant (R2-2bPT) of the gene-silencing suppressor 2b presented mild pathogenicity without stunting, and low RNA accumulation was noticed (Figure 4). When the mutants with TLS mutation in RNA1 (R1-T_m1_ or R1-T_m2_) and R2-2bPT were co-infected, the mutant in RNA1 was repaired at a rate of 100% in progeny RNAs from systemic leaves (Figure 4B), while TLS mutation in RNA1 failed to be repaired (Figure 1B). Similarly, when the TLS mutants in RNA3 (R3-T_m1_ or R3-T_m2_) and R2-2bPT co-infected plants, mutation was repaired at a rate of 100% in progeny RNAs from systemic leaves (Figure 4B), higher than that of a single TLS mutation (R3-T_m1_ or R3-T_m2_) (Figure 1B). The presence of R2-2bPT remarkably improved the repair frequency of TLS mutations in RNA 1 or RNA3, most likely due to the relatively low RNA accumulation resulting from the defect in gene-silencing suppressor 2b (Figure 4C). To further confirm the relationship between the low dose of CMV genome and the TLS mutation repair, inoculation with the dilution of R1-T_m1_/R2/R3 or R1-T_m2_/R2/R3 was performed (Figure 5). TLS mutation repair in RNA1 was noticed for inoculum at OD_600_ = 1.2 × 10^−4^ (Figure 5A,B), which was the dilution end-point of wild-type CMV (Figure 5C). In addition, wild-type CMV and CMV with debilitating cis-elements had the same levels of dilution end-points, because there was no CMV accumulation for the inoculums of OD_600_ = 1.2 × 10^−5^ or OD_600_ = 1.2 × 10^−6^ (Figure 5). Therefore, the critical threshold point of the viral genome dose around the dilution end-point triggered the mutation repair of the debilitating TLS in RNA1, without having a negative effect on the pathogenicity and RNA accumulation of CMV (Figure 1 and Figure 5).

### 3.4. Cell-to-Cell Movement Was Required for TLS Mutation Repair

Previous studies reported that mutations of essential elements in turnip crinkle virus were not repaired in the protoplast but repaired in plants in a direct or indirect manner [28]. It has been implied that mutation repair may require cell-to-cell movement. To analyze this fact, a cell-to-cell movement-defected mutant R3-M5 was constructed through two Ala substitutions corresponding to positions 549 and 554 in the 3a ORF [23] (Figure 6A,B). Since the site of mutation was not in the IGR region, it did not affect the subgenome. When R2-T_m1_ or R2-T_m2_ and R3-M5 were co-inoculated into plants at OD _600_ = 1.2, mutation repair failed to occur (Figure 6C), whereas single R2-T_m1_ or R2-T_m2_ inoculation with wt R3 presented a certain extent of mutation repair at 46% or 35% (Figure 1B and Figure 6C). It is suggested that the defect of cell-to-cell movement blocked the mutation repair at a high dose of inoculum. In addition, the effects of the cell-to-cell movement defect on mutation repair were also analyzed in the dilution inoculation assay (Figure 7). When R1-T_m1_ or R1-T_m2_ and R3-M5 were co-inoculated into plants, mutation repair failed to occur at any concentration of the original dose (Figure 7), whereas single R1-T_m1_ or R1-T_m2_ inoculation near the dilution end-point presented the occurrence of mutation repair (Figure 5). Taken together, defected cell-to-cell movement of the virus can block the mutation repair regardless of a high or low concentration of inoculum.

Taken together, the mutation repair of the TLS mutation in CMV had multiple trigger factors and required the cell-to-cell movement of the virus (Figure 8). Factors included the mutation characteristics of the TLS, the quantity or quality defects of the genome segments, and the quantity defects of the whole genome, particularly around the dilution end-point. Cell-to-cell movement is an obligatory requirement for mutation repair, which is a special case of Muller’s ratchet (Figure 8). Mutation repair is dynamic and based on the fitness change in viruses caused by mutations in functional proteins, functional cis-elements, or the low inoculum doses. In addition, mutation repair relies on fitness changes and cell-to-cell movements in plants.

## 4. Discussion

### 4.1. Similar TLS Elements in Different RNAs of CMV Play Different Roles in CMV Pathogenicity

TLS is essential for the in vitro replication of CMV [21]. The importance of the TLS in in vivo RNA synthesis and the pathogenicity of CMV and other viruses has been reported [29,30,31,32]. TLSs are conserved in CMV RNAs; however, their role in the in vivo synthesis of RNA1, RNA2, and RNA3 was unexplored. In this study, identical TLS mutations in RNA1, RNA2, and RNA3 had remarkably different effects on the synthesis of the corresponding RNAs and different frequencies of mutation repair. The TLS mutation in RNA1 did not affect the synthesis of RNA1 and failed to be repaired at a high inoculum dose. The TLS mutation in RNA2 caused a reduction in RNA2 synthesis and had a high frequency of mutation repair. The TLS mutation in RNA3 increased the size of RNA3 and resulted in the highest frequency of mutation repair (Figure 1). The present study indicates that the TLSs of RNA1, RNA2, and RNA3 play different roles in the synthesis of the corresponding RNAs. Probably, the TLS interacts with other cis-elements present in the CMV RNAs to affect the genomic RNA synthesis in vivo. In addition, TLS mutants in different RNA fragments presented different effects on viral pathogenicity. The TLS mutant in RNA1 or RNA3 still showed severe pathogenicity, while the TLS mutant in RNA2 showed mild pathogenicity. For the TLS mutation in RNA1, the effect of TLS mutation on replication may have been complimented by other potential elements. For TLS mutation in RNA3, the effect of TLS mutation on replication should be complimented by the newly extended sequences, as shown in Appendix A. The extended sequence contains a complete TLS structure sequence. The effect of the TLSs in different RNAs on CMV pathogenicity in vivo was variable, which is more comprehensive than that in vitro. Taken together, similar TLS elements in different CMV RNAs play different roles in the pathogenicity of CMV.

### 4.2. Multiple Levels of Factors Can Trigger Mutation Repair

The error-prone performance of viral RdRp has been shown to cause genetic mutations such as point mutations, insertions, deletions, and replacements [33,34,35]. In addition, other adverse factors and genetic engineering can induce mutations in the viral genome [2]. These natural or artificial mutations can have neutral, debilitating, and lethal effects on the virus genome. Viruses have evolved a number of mechanisms including the negative selection of less-fit variants and the rehabilitation of mutations, directly or indirectly [3]. The latter relies on the infidelity of the viral RdRp through RNA recombination or complementation by other coinfecting viruses [2,13,36]. Based on the data in this study and previous reports, mutation repair can be related to three levels of trigger factors. Firstly, mutation occurs on essential cis-elements, such as TLS, and can be repaired as observed in this study along with previous reports [37,38,39,40]. Secondly, mutation causes quantity or quality defects in genome segments, as in the case of RNA2 or RNA3 in CMV. Finally, the genome is mutated at a low inoculum dose around the dilution end-points. All these factors may cause low fitness of the virus and trigger mutation repair. In addition, a large size change of RNA3 and RNA4 may also be a type of indirect or compensatory mutation repair for TLS mutations (Figure 1 and Figure 2).

### 4.3. Cell-to-Cell Movement Is Obligatory for Mutation Repair

In addition to these trigger factors, TLS mutation repair requires cell-to-cell movement, a bottleneck for this process. Previous studies have shown that bottleneck events are generally disadvantageous [41,42,43,44,45,46] and reduce genetic variation and fitness in the virus population, a phenomenon known as Muller’s ratchet [47,48,49,50,51,52,53]. The possible positive effect of bottlenecks is the removal of defective interfering viruses and the selection of evolutionary trajectories in rugged fitness landscapes [54,55,56,57,58]. In this study, the obligated effect of cell-to-cell movement on mutation repair provides new insight into Muller’s ratchet. In single-cell systems, such as protoplasts, mutation repair failed to occur [28], which implies the indispensable effect of cell-to-cell movement on mutation repair. Therefore, the effect of bottlenecks from cells on virus evolution may have two perspectives. For well-adapted or wild-type viruses, bottlenecks are disadvantageous and result in a reduction in genetic variation and fitness after overcoming barriers, which is the most common case in Muller’s ratchet. For weakly adapted or debilitating viruses, bottlenecks are advantageous, which is shown by the obligated requirement of cell-to-cell movement for mutation repair in this study. Through the cell barrier, the TLS mutation tends to be repaired, and virus fitness is improved. Therefore, bottlenecks may have different effects on viruses with different fitness levels. The detailed mechanism of cell-to-cell movement involved in TLS mutation repair is yet to be elucidated.

### 4.4. Revelation on Construction and the Application of Mild Vaccines from Mutation Repair

Mild strain cross protection is an effective strategy to manage virus diseases. Mild vaccine strains can be acquired from natural mild strains or artificial mild mutants [59]. The major characteristic of a mild vaccine strain is the stability of mild pathogenicity [59,60,61]. However, mutation repair could threaten the safety of mild vaccine strains [2].

Based on results in this study, a low concentration of inoculum can remarkably promote the frequency of mutation repair. Thus, the low concentration of inoculum should be avoided during the application of mild vaccines to depress the low dose-associated mutation repair. In order to analyze the threat of a low dose to the safety of mild vaccines, two types of CMV-attenuated vaccines, R2-2bPT-1 and R2-2bPT-2, were tested in a dilution inoculation assay (Appendix A). Mutant R2-2bPT-1 or R2-2bPT-2 co-inoculation with wt R1 and R3 presented mild pathogenicity (Appendix A). The pre-inoculation of mutant R2-2bPT-1 or R2-2bPT-2 could protect plants from wild-type CMV infection (Appendix A). In the dilution inoculation assay, the mutation repair of R2-2bPT-1 and R2-2bPT-2 occurred at OD_600_ = 1.2 × 10^−4^ and OD_600_ = 1.2 × 10^−5^, respectively (Appendix A). It is suggested that the low concentration of the mild vaccine had the risk of mild pathogenicity to severe pathogenicity. In addition, vector transmission characteristics of mild vaccines should be removed, because the infection of vaccines mediated by vectors such as insects may represent another type of low dose for a mild vaccine. Therefore, the vector transmission characteristics of mild vaccines should be eliminated to avoid the potential mutation repair of mild vaccines. This study provides important revelations on the creation and application of mild vaccines based on RNA viruses.

## 5. Conclusions

Due to the low fidelity of replicase, debilitating RNA viruses can be repaired through different mechanisms, which implies the resilience of RNA viruses. In this study, we identified multiple levels of triggered factors and the occurrence occasions of mutation repair using the divided genome of CMV, which contains a conserved cis-element tRNA-like structure (TLS) at the 3′end. The TLS mutation of different RNAs in CMV presented different rates of mutation repair from 0 to 80%. TLS mutation—resulting in genomic quality or quantity defects, or low doses around the dilution end-points—was inclined to be repaired. However, all the above types of mutation repair required viral cell-to-cell movement, presenting the positive effect of cell-to-cell bottlenecks on virus evolution and the increased fitness of low-fitness RNA viruses. A special phenomenon in the Muller ratchet is that this bottleneck often decreases the fitness of viruses.

## Figures and Tables

**Figure 1 biology-11-01051-f001:**
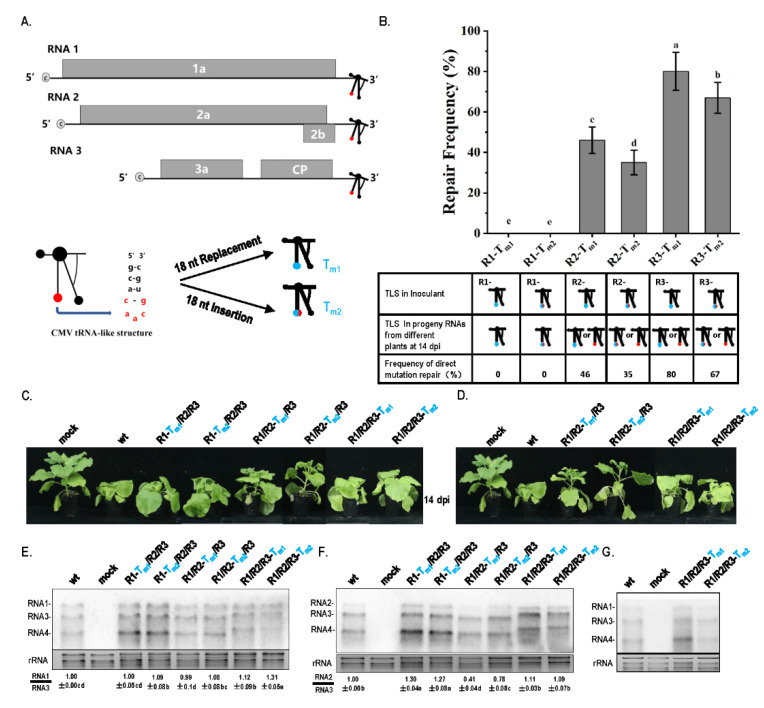
The repair frequency and pathogenicity of TLS mutation in different CMV RNAs. (**A**) Schematic representation of CMV and mutation on TLS. TLS is indicated by special branch structure, and essential caacg loop is indicated by solid red circle. Mutation of 18 nt replacement on caacg (T_m1_) or 18 nt insertion before caacg (T_m2_) is indicated by solid blue circle. (**B**) Frequency of repair of TLS mutation in RNAs of CMV_Fny_. R1—RNA1 of CMV_Fny_; R2—RNA2 of CMV_Fny_; R3—RNA3 of CMV_Fny_. (**C**) Pathogenicity of different TLS mutants without mutation repair: wt—RNA1/RNA2/RNA3 of wild-type CMV_Fny_. (**D**) Pathogenicity of different TLS mutants with mutation repair. (**E**) Northern blot for RNA samples without direct repair of TLS mutation, using probes against RNA1, RNA3, and RNA4. (**F**) Northern blot for RNA samples without direct repair of TLS mutation, using probes against RNA2, RNA3, and RNA4. The blot signals were analyzed with Quantity One software. (**G**) Northern blot for RNA samples with TLS mutation repair, using probes against RNA1, RNA3, and RNA4. The data were analyzed with independent sample LSD tests using DPS software, different letters indicate that values of the two treatments were significantly different at *p* < 0.05.

**Figure 2 biology-11-01051-f002:**
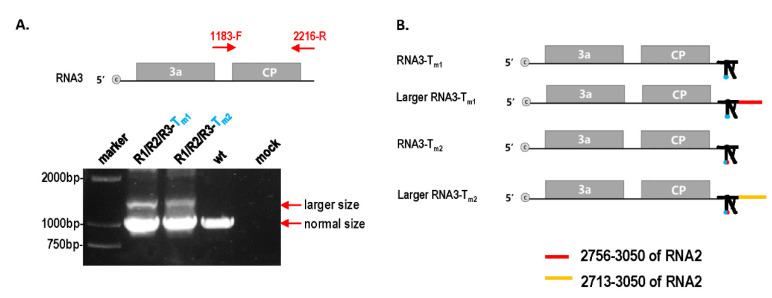
Identification of large size RNA3 in R1/R2/R3-T_m1_ and R1/R2/R3-T_m2_. (**A**) RT-PCR of position 1183–2216 in RNA3; 1183-F and 2216-R indicate the forward and reverse primers. (**B**) Schematic diagram of large RNA3 in R1/R2/R3-T_m1_ and R1/R2/R3-T_m2_.

**Figure 3 biology-11-01051-f003:**
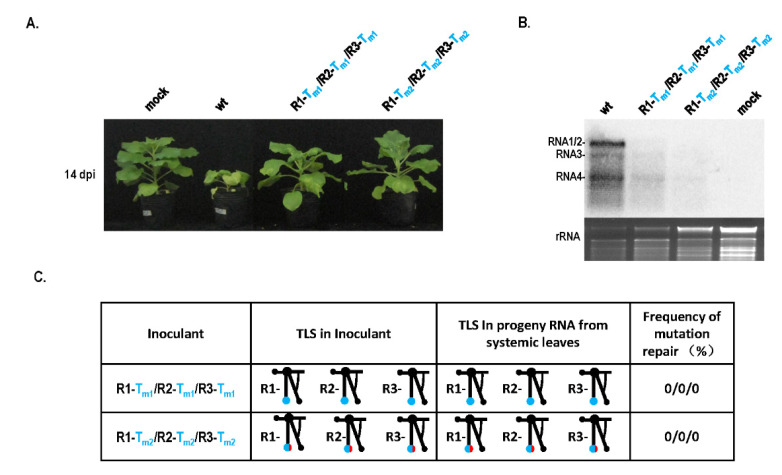
Pathogenicity and mutation repair by mixed infection of TLS mutants in CMV RNAs. (**A**) Pathogenicity of mixed infection of TLS mutants in CMV RNAs. (**B**) Northern blot of RNA samples for mixed infection of TLS mutation. (**C**) Mutation repair analysis of mixed infection of TLS mutants in different RNAs of CMV.

**Figure 4 biology-11-01051-f004:**
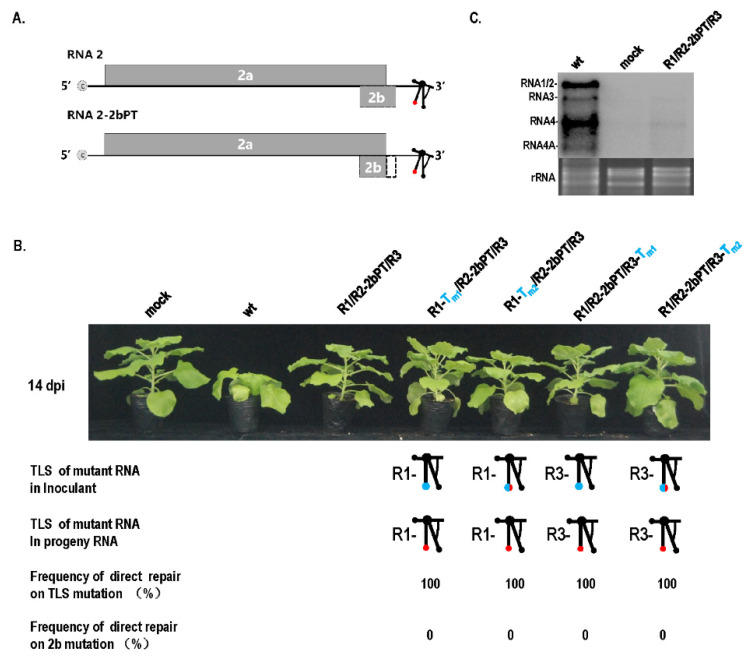
Mutation repair of the TLS mutation of RNA1 or RNA3 in the presence of 2b pre-termination in RNA2. (**A**) Schematics of 2b pre-termination mutant in RNA2. The dotted area indicates the partial deletion of partial 2b ORF caused by the pre-termination of 2b. PT—pre-termination. (**B**) Mutation repair of the TLS mutation of RNA1 or RNA3 in the presence of 2b pre-terminated RNA2. (**C**) Effect of 2b pre-termination on the RNA accumulation of CMV. wt—RNA1/RNA2/RNA3 of wild-type CMV_Fny_. R2-2bPT—mutant of 2b resulted from pre-termination.

**Figure 5 biology-11-01051-f005:**
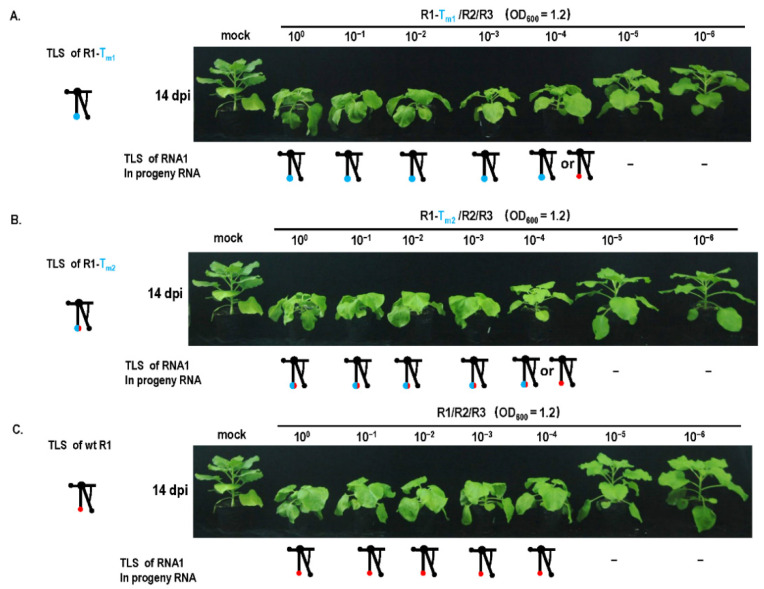
Dilution inoculation of CMV_Fny_ or its variants and mutant repair assay. (**A**) Mutation repair assay under dilution inoculation of R1-T_m1_/R2/R3. R1—RNA1 of CMV_Fny_; R2—RNA2 of CMV_Fny_; R3—RNA3 of CMV_Fny_; −—no detectable viral RNA. (**B**) Mutation repair assay under dilution inoculation of R1-T_m2_/R2/R3. (**C**) Mutation repair assay under dilution inoculation of CMV_Fny_.

**Figure 6 biology-11-01051-f006:**
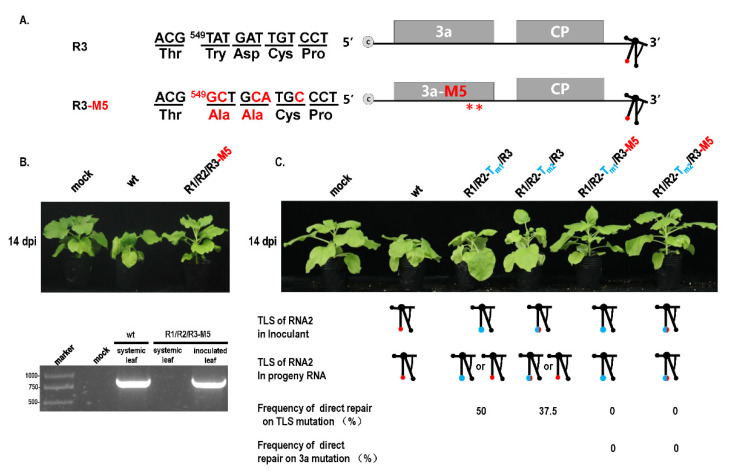
Repair assay of TLS mutation in RNA2 with movement-defected CMV. (**A**) Mutation on movement protein 3a in CMV. Mutation of nucleotide and amino acid is marked in red. **—Mutation at two amino acids in 3a protein. (**B**) Effect of 3a mutation on pathogenicity of CMV. (**C**) Effect of 3a mutation on repair of TLS mutation in RNA2.

**Figure 7 biology-11-01051-f007:**
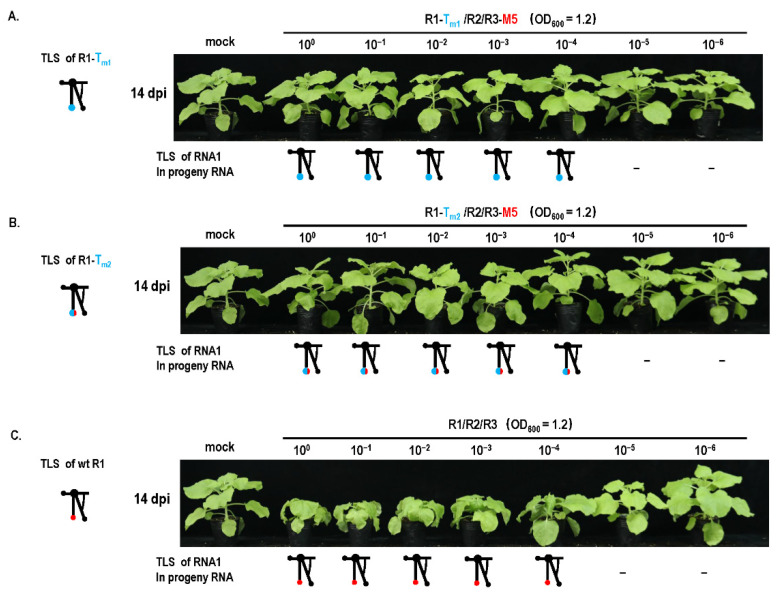
Effect of cell-to-cell movement on mutant repair in dilution inoculation assay. Mutation repair assay under dilution inoculation of (**A**) R1-T_m1_/R2/R3-M5. R1—RNA1 of CMV_Fny_; R2—RNA2 of CMV_Fny_; R3-M5—3a mutation in RNA3 of CMV_Fny_. (**B**) R1-T_m2_/R2/R3-M5. (**C**) Wild-type CMV.

**Figure 8 biology-11-01051-f008:**
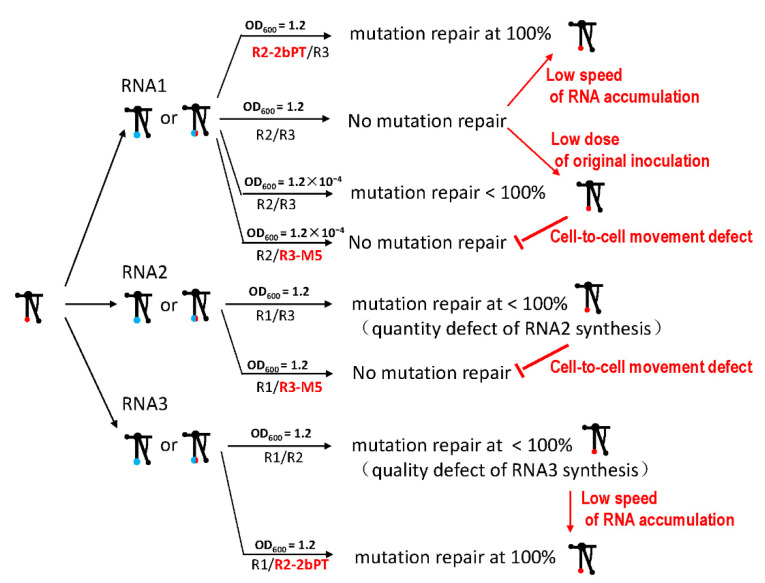
Summary of mutation repair of TLS mutation in different RNAs of CMV. Red arrow indicates the transition from no or low mutation repair to the high mutation repair; red T-shape indicates the transition from the presence of mutation repair to the absence of mutation repair. The words next to the arrow are the regulatory factors.

## Data Availability

The relevant datasets supporting the results of this article are included within the article and its additional files.

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
