# Peer review of "Multiple Levels of Triggered Factors and the Obligated Requirement of Cell-to-Cell Movement in the Mutation Repair of Cucumber Mosaic Virus with Defects in the tRNA-like Structure"

_biology, 2022, doi:10.3390/biology11071051_

Round 1
Reviewer 1 Report
Authors of the manuscript titled "Multiple levels of triggered factors and obligated requirement of cell-to-cell movement in mutation repair of cucumber mosaic virus with defect of the tRNA-like structure" submitted to MDPI journal Biology have improved the manuscript based on the suggestions to a major extent compared to the previous version. The manuscript has addressed all suggestions and incorporated all corrections suggested previously. The manuscript can be accepted for publication.
Author Response
Dear reviewer:
Thanks for your comments. According to your comments on the English language and suggestions from other reviewers, we have made slight revisions to the manuscript, including abstract, introduction, discussion and some annotation of figures. Please check the new version.

Reviewer 2 Report
This study aimed to investigate the role of RNA recombination in repair of debilitating mutations in RNA virus genomes and the factors contributing to such restoration. The experimental system used involved manipulating the full-length cDNA clones of a three-component Cucumber mosaic virus (CMV) to introduce mutations in the 3' terminal tRNA-like structures (TLS) of CMV RNAs. Two types of 18 nt-long insertions were introduced to the TLS sequences of each of three genomic RNAs of CMV. Also the RNA2 construct with a mutation in the 2b RNAi suppressor gene (leading to the expression of truncated non-functional 2b antiviral RNAi protein and mild viral infection), and the RNA3 version with mutation in the the cell-to-cell 3a movement protein gene (abolishing cell-to-cell movement of the virus) were designed. N. benthamiana plants were Agroinoculated with different combinations and dilutions of the mutated and wild-type constructs to initiate infection. Virus infection was monitored in the systemic, non-infiltrated leaves and included observation of symptoms and sequencing of the RT-PCR fragments corresponding to the 3' terminal regions of each of three CMV RNAs to determine if TLS were repaired.
It was found that in the case of co-infiltration of the plants with the mixture of RNA1+2+3 each containing mutated TLS, systemic leaves did not contain progeny with the repaired TLS (Fig. 3). But when only one RNA component contained TLS mutation and the other two contained wild-type TLS, recombinational repair took place. High TLS repair rates were observed in al cases when RNA2 and RNA3 were Agroinfiltrated with other wild-type components. Interestingly, RNA1 TLS showed a variation of the repair rates, ranging from 0% (in the case of "standard" high-dose infiltration with wild-type RNA2 and RNA3), to 100% in the case of mild virus infection caused by the truncation of the 2b RNAi suppressor (R2-2b). Some restoration of the RNA1 TLS occurred in the case of highly diluted infection with wild-type RNA2 and RNA3. No recombination repair of TLS was observed in the case of the 3a mutant (R3-M5) variant which was not able to move cell-to-cell. It was suggested that the bottleneck caused by either reduced replication in the case of mild (truncated 2b) variant, low doses in the case of high dilution, or in the course of cell-to-cell virus movement could promote recombination repair.
The revised version shows much improvement compared to the original submitted in the end of 2021. In particular, new version had investigated why RNA3 and RNA4 in the restored virus progeny of R1/R2/R-Tm1 and R1/R2/R3-Tm2. It was found that a sections of RNA2 were included to the RNA3 / RNA4 (new Figure 2).
This study is very interesting and provides a novel data on the the role of RNA recombination in RNA virus biology, in particular, on the role of infection severity and cell-to-cell movement in the recombinational repair processes, but many points require clarification:
L84-85/ Fig. 1. The MS cited previous study (ref 21) which indicates that TLS elements are essential for CMV RNAs replication (L84-85). But this contradicts the results presumed in Fig. 1 B, which shows that CMV remained infectious when RNA1 had either Tm1 or Tm2 mutations which were not restored to wild-type (Fig. 1B). This contradiction needs clarification and discussion.
L 24-25. Re-writer the sentence on mutation correction of RNA2 and RNA3. It is not clear in ints present form. What are the "relative quantity defects" and "quality defects" ?
L 27. "2b" -> "2b gene", also specify that 2b is an RNAi suppressor.
L 37. " negative selection " - please specify ...
L 71-72. Prove a reference for the statement that RNA recombination doesn't require complicated machinery.
L 87. "The size of genome" ?
L 88" (the quantity of genome replication" ? Do you man number of replication cycles.
Author Response
Dear reviewer:
Thanks for your comments. We have made revisions to the manuscript according to your comments, including abstract, introduction, discussion and some annotation of figures. Please see the attachment.

This manuscript is a resubmission of an earlier submission. The following is a list of the peer review reports and author responses from that submission.
Round 1
Reviewer 1 Report
Liu et al submitted an article titled, ‘Multiple levels of triggered factors and positive effect of cell-2 to-cell bottleneck in mutation repair of RNA viruses’ to the MDPI Biology.
The article is very long, well written, and the work is interesting. The manuscript needs few minor modifications.
Few sentences are very long (eg. Line 81). It is important to keep sentences short, if possible, in order to keep the reader on track, and also to minimize errors.
Author Response
Liu et al submitted an article titled, ‘Multiple levels of triggered factors and positive effect of cell-2 to-cell bottleneck in mutation repair of RNA viruses’ to the MDPI Biology.
The article is very long, well written, and the work is interesting. The manuscript needs few minor modifications.
Few sentences are very long (eg. Line 81). It is important to keep sentences short, if possible, in order to keep the reader on track, and also to minimize errors.
Answer: Thanks for your suggestion. I have modified sentences which seem to be long. In addition, we also made major revision based on comments of other reviewers. Please check the revised version.
Reviewer 2 Report
The authors of the manuscript "Multiple levels of triggered factors and positive effect of cell-to-cell bottleneck in mutation repair of RNA viruses'', have explored the factors that can repair the mutation in the viral genome that helps in maintaining the evolution of viral genomes in spite of adverse environmental conditions. RNA recombination is one of the tools used by viruses to facilitate mutation repair. The authors here have used CMV as a model to study the mutation repair mediated by the RNA recombination. The RNA, RNA2, and RNA3 in CMVhave identical 3' UTR containing TLS (t-RNA like structure), which was essential for replication. Mutations in the TLS of the 3 RNAs had a different extent of mutation repair by RNA recombination.
The authors methodically have supported their hypothesis that the extent of mutation repair by RNA recombination is dependent on several factors, which in turn define the pathogenicity of the virus. The authors have also experimented on the cell-to-cell movement for mutation repair to be important for this positive effect. On a futuristic note, this manuscript highlights the fact that while creating mild vaccines mutation in TLS should be preferably avoided, as they are prone to mutation repair by RNA recombination.
Author Response
The authors of the manuscript "Multiple levels of triggered factors and positive effect of cell-to-cell bottleneck in mutation repair of RNA viruses'', have explored the factors that can repair the mutation in the viral genome that helps in maintaining the evolution of viral genomes in spite of adverse environmental conditions. RNA recombination is one of the tools used by viruses to facilitate mutation repair. The authors here have used CMV as a model to study the mutation repair mediated by the RNA recombination. The RNA, RNA2, and RNA3 in CMV have identical 3' UTR containing TLS (t-RNA like structure), which was essential for replication. Mutations in the TLS of the 3 RNAs had a different extent of mutation repair by RNA recombination.
The authors methodically have supported their hypothesis that the extent of mutation repair by RNA recombination is dependent on several factors, which in turn define the pathogenicity of the virus. The authors have also experimented on the cell-to-cell movement for mutation repair to be important for this positive effect. On a futuristic note, this manuscript highlights the fact that while creating mild vaccines mutation in TLS should be preferably avoided, as they are prone to mutation repair by RNA recombination.
Answer: Thanks for your evaluation on this manuscript. Based on this manuscript, several factors (characteristic change of cis-elements, low original concentration of mutant, and cell-to-cell movement) were identified to be involved in the mutation repair. It provided on application of mild vaccine: avoid of the low dose of original inoculation and removal of vector-transmission characteristic. In addition, TLS was not good target to create mild mutation due to the potential mutation repair.
In addition, we also made major revision based on comments of other reviewers. Please check the revised version.
Reviewer 3 Report
Liu et al. attempt to analyze how recombination may affect the repair of the genome of a plant virus (cucumber mosaic virus, CMV). Although the subject of the manuscript is relevant to understand virus evolution and replication, the work has may important flaws. First, material and methods are poorly described, such that reproducibility is highly compromised; and it is unclear whether the experimental design shown is appropriate to answer the questions posed. Second, results are supported by no statistical analyses, and in some parts are incorrectly interpreted. Third, many of the conclusions are based on observational comparisons and, as I just mentioned, the lack of statistics makes these conclusions doubtful. I list specific comments below:
- Introduction: Some aspects of the conceptual background appear repetitively in this section. For instance, the benefits of recombination are listed at least three times. On the other hand, there is no single explanation on the experimental system chosen. As not every reader will be familiar with CMV, some description of genomic structure and protein function should be included. Also, please use the correct terminology: CMV has not a “divided” genome, it is a *multipartite virus* (note that viruses with divided genomes can be segmented or multipartite, which is not the same).
- I wonder whether the experimental design is appropriate for the question posed. The authors say in line 320 that polymerase errors may lead to indels. Thus, repairing of Tm2 mutants does not necessarily involve recombination, and can be generated just by polymerase slippage. This is not contemplated at all.
- There is a general lack of data in the Results sections. For instance, it is not specified on how many plants are based the percentages presented. Were all repeated experiments pooled together? If the authors did that, they should have tested first variability between experiments. Otherwise, data cannot be pooled. Also, it is unclear whether repaired and unrepaired variants were detected in the same plant or if the presence of one completely displaced the other. Moreover, saying that 37% is lower than 50% is quite bold given that not even a chi-square test is presented. Given the low number of plants that appear to be used in the experiments (as this number is not shown), I doubt that any statistical test would yield significant differences between these figures. Hence, all conclusions based on these analyses are very weak.
- The figures showing symptoms are really bad. It is not possible so see anything there, and this reviewer cannot compare virus pathogenicity just based on plant height. Also, it is unclear why the panels with plants showing symptoms are not comparable. For instance, in Figure 1 panels C and D do not have the same treatments. If the authors state that symptoms were the same in R1 mutants, they should demonstrate that. Moreover, these pictures show an example of a single plant, however, in many cases repair percentages are not 100%: did plants with and without repair express different symptoms in the same treatment?
- The authors repeatedly state that R2 mutants have less RNA2. Less than what? According to panel F of Figure 1, these mutants have apparently (again, no quantification is shown) the same RNA 2 than the wildtype. Hence, if any, the conclusion would be that R1 and R3 mutants increase R2 concentration, which invalidates one of the main conclusions of the work and break the claimed correlation between repair and RNA level.
- Another quite surprising result is that in R3 mutants RNAs3 and 4 change in size more than should be expected given the introduced mutations. However, this is just overlooked and no further analyses on this phenomenon are presented: where these mutants sequenced? What could account for this change of size? How would this extra genomic material affect the performance of the mutants? Note that the presence of this unknow material do not allow to directly link the presence of mutations ad the phenotypes observed.
- Lines 167-177. I think copy-choice is possible in Tm2 mutants as the template is there (the wildtype sequence is not removed). Thus, the conclusion is wrong.
- Discussion: Effect of the lack of cell-to-cell movement on RNA repair may have nothing to do with bottlenecks here. Perhaps is just a matter of the number of replication rounds needed for a copy-choice event to occur as this is an infrequent event. It is well possible that in mutants impaired in movement the virus is limited to a few cells and the number of replication rounds is not enough to generate and amplify the repaired genome. Also, the authors do not demonstrate presence or absence of systemic movement in the mutants, nor the existence of bottlenecks (a concept that is in the title of the work!) so the reader is let to assume that this is so.
Author Response
Liu et al. attempt to analyze how recombination may affect the repair of the genome of a plant virus (cucumber mosaic virus, CMV). Although the subject of the manuscript is relevant to understand virus evolution and replication, the work has may important flaws. First, material and methods are poorly described, such that reproducibility is highly compromised; and it is unclear whether the experimental design shown is appropriate to answer the questions posed. Second, results are supported by no statistical analyses, and in some parts are incorrectly interpreted. Third, many of the conclusions are based on observational comparisons and, as I just mentioned, the lack of statistics makes these conclusions doubtful. I list specific comments below:
- Introduction: Some aspects of the conceptual background appear repetitively in this section. For instance, the benefits of recombination are listed at least three times. On the other hand, there is no single explanation on the experimental system chosen. As not every reader will be familiar with CMV, some description of genomic structure and protein function should be included. Also, please use the correct terminology: CMV has not a “divided” genome, it is a *multipartite virus* (note that viruses with divided genomes can be segmented or multipartite, which is not the same).
Answer: Thanks for your suggestion. I have modified the introduction.
I wonder whether the experimental design is appropriate for the question posed. The authors say in line 320 that polymerase errors may lead to indels. Thus, repairing of Tm2 mutants does not necessarily involve recombination, and can be generated just by polymerase slippage. This is not contemplated at all.
Answer: Yes. Error-prone performance of the viral RdRp has two-sided function:inducing mutations and mutation repair. So we test the possible mechanism of mutation repair on these TLS mutation in this study. In new figure 3 (the previous Supplement Figure 1), when R1-Tm1/R2-Tm1/R3-Tm1 or R1-Tm2/R2-Tm2/R3-Tm2 were used to inoculate plants, mutation repair failed to occur. It is suggested that repairing of TLS mutants does not generated by RdRp slippage but by RNA recombination. Repair of TLS mutation required recombination template of wt TLS.
- There is a general lack of data in the Results sections. For instance, it is not specified on how many plants are based the percentages presented. Were all repeated experiments pooled together? If the authors did that, they should have tested first variability between experiments. Otherwise, data cannot be pooled. Also, it is unclear whether repaired and unrepaired variants were detected in the same plant or if the presence of one completely displaced the other. Moreover, saying that 37% is lower than 50% is quite bold given that not even a chi-square test is presented. Given the low number of plants that appear to be used in the experiments (as this number is not shown), I doubt that any statistical test would yield significant differences between these figures. Hence, all conclusions based on these analyses are very weak.
- Answer: For each treatment, 10-20 plants was tested in at least three batch of experiments. Repaired or unrepaired variants of TLS mutation can not be detected in the same plant and presented exclusive characteristic. Based on these, we get the frequency of mutation repair.
In this part, we have modified and add statistical test. Please check the revised version.
- The figures showing symptoms are really bad. It is not possible so see anything there, and this reviewer cannot compare virus pathogenicity just based on plant height. Also, it is unclear why the panels with plants showing symptoms are not comparable. For instance, in Figure 1 panels C and D do not have the same treatments. If the authors state that symptoms were the same in R1 mutants, they should demonstrate that. Moreover, these pictures show an example of a single plant, however, in many cases repair percentages are not 100%: did plants with and without repair express different symptoms in the same treatment?
Answer: In this study, we used the plant height to describe the pathogenicity of CMV variants, because change of plant height is the most remarkable characteristic for pathogenicity. Other symptom such as leafing curling can not be compared directly. In addition, we showed the pathogenicity of CMV variants without TLS mutation repair in Figure 1C and that of CMV variants with TLS mutation repair in Figure 1D. Plants with and without repair express similar symptoms in the same treatment, and it is implied that repaired variant may need a period to show the severe pathogenicity .
- The authors repeatedly state that R2 mutants have less RNA2. Less than what? According to panel F of Figure 1, these mutants have apparently (again, no quantification is shown) the same RNA 2 than the wildtype. Hence, if any, the conclusion would be that R1 and R3 mutants increase R2 concentration, which invalidates one of the main conclusions of the work and break the claimed correlation between repair and RNA level.
Answer: we added the ratio data of RNA2/RNA3 in Fig.1F and also modified the description.
- Another quite surprising result is that in R3 mutants RNAs3 and 4 change in size more than should be expected given the introduced mutations. However, this is just overlooked and no further analyses on this phenomenon are presented: where these mutants sequenced? What could account for this change of size? How would this extra genomic material affect the performance of the mutants? Note that the presence of this unknow material do not allow to directly link the presence of mutations ad the phenotypes observed.
Answer: Yes, this is a quite surprising result. In revised version, we also added another Northern blot in Figure 1G, which is the Northern result for samples with the TLS mutation. In Figure 1G, size change of RNA3 and RNA 4 disappear. It is suggested that size change of RNA3 and RNA4 was correlated with TLS mutation in RNA3.
- Such increases of RNA3 / RNA4 occur in the case of all plants without direct repair of TLS mutation. We also performed RT-PCR to identify the extended sequences, which derived from the 3’end of RNA2 shown as new Figure 2.
- Lines 167-177. I think copy-choice is possible in Tm2 mutants as the template is there (the wildtype sequence is not removed). Thus, the conclusion is wrong.
Answer: Although the wildtype caacg sequence is not removed in Tm2 mutants, no mutation repair occurred when RNA1, RNA2 and RNA3 simultaneously contained Tm2 mutation. The occur of copy-choice may require the pure wildtype sequence, after all caacg in Tm2 mutation was mixed with insertion sequences. About this section, I modified the description. Please check it.
- Discussion: Effect of the lack of cell-to-cell movement on RNA repair may have nothing to do with bottlenecks here. Perhaps is just a matter of the number of replication rounds needed for a copy-choice event to occur as this is an infrequent event. It is well possible that in mutants impaired in movement the virus is limited to a few cells and the number of replication rounds is not enough to generate and amplify the repaired genome. Also, the authors do not demonstrate presence or absence of systemic movement in the mutants, nor the existence of bottlenecks (a concept that is in the title of the work!) so the reader is let to assume that this is so.
Answer: Yes, your suggestion is right. I removed the description on bottlenecks. In addition to the effect of cell-to-cell movement on the mutation repair in this study, we also mentioned the case in protoplast in which no mutation repair occurred. All of these data implied the obligate requirement of cell-to-cell movement on mutation repair, although detailed mechanism was not clear.
Reviewer 4 Report
This study aimed to investigate the role of RNA recombination in repair of debilitating mutations in RNA virus genomes and the factors contributing to such restoration. The experimental system used involved manipulating the full-length cDNA clones of a three-component Cucumber mosaic virus (CMV) to introduce mutations in the 3' terminal tRNA-like structures (TLS) of CMV RNAs. Two types of 18 nt-long insertions were introduced to the TLS sequences of each of three genomic RNAs of CMV. Also the RNA2 construct with a mutation in the 2b RNAi suppressor gene (leading to the expression of truncated non-functional 2b protein and mild viral infection), and the RNA3 version with mutation in the the cell-to-cell 3a movement protein gene (abolishing cell-to-cell movement of the virus) were designed. N. benthamiana plants were Agroinoculated with different combinations and dilutions of the mutated and wild-type constructs to initiate infection. Virus infection was monitored in the systemic, non-infiltrated leaves and included observation of symptoms and sequencing to the RT-PCT fragments corresponding to the 3' terminal regions of each of three CMV RNAs to determine if TLS were repaired.
It was found that in the case of co-infiltration of the plants with the mixture of RNA1+2+3 each containing mutated TLS, systemic leaves did not contain progeny with the repaired TLS. But when only one RNA component contained TLS mutation and the other two contained wild-type TLS, recombinational repair took place. High TLS repair rates were observed in al cases when RNA2 and RNA3 were Agroinfiltrated with other wild-type components. Interestingly, RNA1 TLS showed a variation of the repair rates, ranging from 0% (in the case of "standard" high-dose infiltration with wild-type RNA2 and RNA3), to 100% in the case of mild virus infection caused by the truncation of the 2b RNAi suppressor (R2-2b). Some restoration of the RNA1 TLS occurred in the case of highly diluted infection with wild-type RNA2 and RNA3. No recombination repair of TLS was observed in the case of the 3a mutant (R3-M5) variant which was not able to move cell-to-cell. It was suggested that the bottleneck caused by either reduced replication in the case of mild (truncated 2b) variant, low doses in the case of high dilution, or in the course of cell-to-cell virus movement could promote recombination repair.
In general, this study is very interesting and provided a novel insight into the role of RNA recombination in RNA virus biology, as well as the the role of infection severity and cell-to-cell movement in the recombinational repair processes, but several points should be addressed:
L158-159 (Fig. 1B) /L.242-243 (Fig. 3A.B). There is a contradiction between statements on the possibility of the repair of the mutated TLS in RNA1. Figure 3 (L242-243) shows that the RNA1 TLS could be repaired (in the context of the mild "truncated 2b" infection or in the case of high dilution. Therefore the statement that "TLS in RNA1 ... did not appear mutation repair" (L.158-159) is not correct and misleading and should be corrected.
L174-177. "it was suggested that repair of TLS was mediated by copy-voice type of RNA recombination". Was it possible to determine by sequencing of the RT-PCR fragments (L123-128) from which wild-type RNA they were coming from (for example, for Figure 2, if TLS in the repaired RNA1 originated from RNA2 (R2-2b) or RNA3).
L.98-100/ L. 153-154 / Figure 1. Include the alignment of the tRNA-like structures (TLS) of CMV RNA1, RNA2 and RNA3 (wild-type and with Tm1 and Tm2 mutations) to Figure 1 or as a separate figure. Also, a secondary structure of CMV TLS could be included to explain the schematic representations of TLS.
L170-172/ Supplementary Figure 1. It is not clear if "R1-Tm1/R2-Tm1/R3-Tm1" or "R1-Tm2/R2-Tm2/R3-Tm2" is infectious. Supplementary Figure 1 should include Northern blot showing the development of CMV infection and this modified Figure should be moved to the main text.
L. 213-214/Figure 1 E, F. Northern blot analysis of R1/R2/R-Tm1 and R1/R2/R3-Tm2 showed that RNA3 and RNA4 components were longer than the RNA3 and RNA4 of wild-type and RNA1 and RNA2 mutants. It is very likely that an extended sequence contained insertion(s) into the 3' half of the repaired RNA3. Did such increases of RNA3 / RNA4 occur in the case of all plants where TLS repair took place? To determine the cause of such RNA3 size increases, additional RT-PCT fragments covering entire RNA3 sequence could be produced and sequenced.
L. 10. Connection between low fidelity of the RNA replication (which results in point mutations) and the rate of recombination (which was a subject of this study) is not obvious.
Author Response
This study aimed to investigate the role of RNA recombination in repair of debilitating mutations in RNA virus genomes and the factors contributing to such restoration. The experimental system used involved manipulating the full-length cDNA clones of a three-component Cucumber mosaic virus (CMV) to introduce mutations in the 3' terminal tRNA-like structures (TLS) of CMV RNAs. Two types of 18 nt-long insertions were introduced to the TLS sequences of each of three genomic RNAs of CMV. Also the RNA2 construct with a mutation in the 2b RNAi suppressor gene (leading to the expression of truncated non-functional 2b protein and mild viral infection), and the RNA3 version with mutation in the the cell-to-cell 3a movement protein gene (abolishing cell-to-cell movement of the virus) were designed. N. benthamiana plants were Agroinoculated with different combinations and dilutions of the mutated and wild-type constructs to initiate infection. Virus infection was monitored in the systemic, non-infiltrated leaves and included observation of symptoms and sequencing to the RT-PCT fragments corresponding to the 3' terminal regions of each of three CMV RNAs to determine if TLS were repaired.
It was found that in the case of co-infiltration of the plants with the mixture of RNA1+2+3 each containing mutated TLS, systemic leaves did not contain progeny with the repaired TLS. But when only one RNA component contained TLS mutation and the other two contained wild-type TLS, recombinational repair took place. High TLS repair rates were observed in al cases when RNA2 and RNA3 were Agroinfiltrated with other wild-type components. Interestingly, RNA1 TLS showed a variation of the repair rates, ranging from 0% (in the case of "standard" high-dose infiltration with wild-type RNA2 and RNA3), to 100% in the case of mild virus infection caused by the truncation of the 2b RNAi suppressor (R2-2b). Some restoration of the RNA1 TLS occurred in the case of highly diluted infection with wild-type RNA2 and RNA3. No recombination repair of TLS was observed in the case of the 3a mutant (R3-M5) variant which was not able to move cell-to-cell. It was suggested that the bottleneck caused by either reduced replication in the case of mild (truncated 2b) variant, low doses in the case of high dilution, or in the course of cell-to-cell virus movement could promote recombination repair.
In general, this study is very interesting and provided a novel insight into the role of RNA recombination in RNA virus biology, as well as the the role of infection severity and cell-to-cell movement in the recombinational repair processes, but several points should be addressed:
L158-159 (Fig. 1B) /L.242-243 (Fig. 3A.B). There is a contradiction between statements on the possibility of the repair of the mutated TLS in RNA1. Figure 3 (L242-243) shows that the RNA1 TLS could be repaired (in the context of the mild "truncated 2b" infection or in the case of high dilution. Therefore the statement that "TLS in RNA1 ... did not appear mutation repair" (L.158-159) is not correct and misleading and should be corrected.
Answer: Thanks for your meticulous suggestion. In figure 1, TLS mutation in RNA1 was not repaired. In figure 3, TLS mutation in RNA1 was repaired. In these two assays, the dose of the original inoculation was not same, which suggested that the inoculation dose is one of the triggered factors of mutation repair. As your suggestion, we modified the statement (L158-159) as “At 14dpi of agroinfiltration at OD600=1.2, two type of mutation on TLS in RNA1 (R1-Tm1 and R1-Tm2) still existed in progeny RNAs and did not appear mutation repair, while mutation of TLS in RNA2 or RNA3 presented different frequency of mutation repair in progeny RNAs (Fig.1B)”
L174-177. "it was suggested that repair of TLS was mediated by copy-voice type of RNA recombination". Was it possible to determine by sequencing of the RT-PCR fragments (L123-128) from which wild-type RNA they were coming from (for example, for Figure 2, if TLS in the repaired RNA1 originated from RNA2 (R2-2b) or RNA3).
Answer:All mutation repair of TLS mutation in this study was recovered to wild type TLS sequences. TLS mutation regions in RNA1, RNA2 and RNA3 was conserved, it is impossible to determine the origin of wt TLS. Based on the data of new figure 2 (previous supplementray figure1), no mutation repair occurred if RNA1/RNA2/RNA3 contained same type of TLS mutation, which suggested that mutation repair required wt TLS template from other RNA component of CMV.
L.98-100/ L. 153-154 / Figure 1. Include the alignment of the tRNA-like structures (TLS) of CMV RNA1, RNA2 and RNA3 (wild-type and with Tm1 and Tm2 mutations) to Figure 1 or as a separate figure. Also, a secondary structure of CMV TLS could be included to explain the schematic representations of TLS.
Answer:Good suggestion. We added a new supplementary Figure 1 to align the TLS of CMV RNA1, RNA2 and RNA3 (wild-type and with Tm1 and Tm2 mutations).
L170-172/ Supplementary Figure 1. It is not clear if "R1-Tm1/R2-Tm1/R3-Tm1" or "R1-Tm2/R2-Tm2/R3-Tm2" is infectious. Supplementary Figure 1 should include Northern blot showing the development of CMV infection and this modified Figure should be moved to the main text.
Answer: "R1-Tm1/R2-Tm1/R3-Tm1" or "R1-Tm2/R2-Tm2/R3-Tm2" is infectious, because the progeny RNA was detected from systemic leaves and presented the existence of TLS mutation. As your suggestion, we changed the Supplementary Figure 1 to Figure 2. The previous other Figures were also modified in order. In addition, we also performed Northern blot, which presented the low RNA accumulation of R1-Tm1/R2-Tm1/R3-Tm1 or R1-Tm2/R2-Tm2/R3-Tm2.
- 213-214/Figure 1 E, F. Northern blot analysis of R1/R2/R-Tm1 and R1/R2/R3-Tm2 showed that RNA3 and RNA4 components were longer than the RNA3 and RNA4 of wild-type and RNA1 and RNA2 mutants. It is very likely that an extended sequence contained insertion(s) into the 3' half of the repaired RNA3. Did such increases of RNA3 / RNA4 occur in the case of all plants where TLS repair took place? To determine the cause of such RNA3 size increases, additional RT-PCT fragments covering entire RNA3 sequence could be produced and sequenced.
Answer: Such increases of RNA3 / RNA4 occur in the case of all plants without direct repair of TLS mutation. Good suggestion. We also performed RT-PCR to identify the extended sequences in larger RNA3, which is derived from 3’end of RNA2 (New Figure 2).
- 10. Connection between low fidelity of the RNA replication (which results in point mutations) and the rate of recombination (which was a subject of this study) is not obvious.
Answer: Yes, you are right. We re-write the simple summary and introduction. Please check them.